# Kuru, the First Human Prion Disease [note 1]

**DOI:** 10.3390/v11030232

**Published:** 2019-03-07

**Authors:** Paweł P. Liberski, Agata Gajos, Beata Sikorska, Shirley Lindenbaum

**Affiliations:** 1Laboratory of Electron Microscopy and Neuropathology, Department of Molecular Pathology and Neuropathology, Medical University Lodz, 90-419 Lodz, Poland; elmo@csk.umed.lodz.pl; 2Department of Extrapyramidal Diseases, Medical University of Lodz, 90-419 Lodz, Poland; agata.gajos@umed.lodz.pl; 3Department of Anthropology, Graduate Center, City University of New York, New York, NY 10016, USA; shirleylindenbaum@gmail.com

**Keywords:** Kuru, prion diseases, neuropathology, Carleton Gajdusek

## Abstract

Kuru, the first human prion disease was transmitted to chimpanzees by D. Carleton Gajdusek (1923–2008). In this review, we summarize the history of this seminal discovery, its anthropological background, epidemiology, clinical picture, neuropathology, and molecular genetics. We provide descriptions of electron microscopy and confocal microscopy of kuru amyloid plaques retrieved from a paraffin-embedded block of an old kuru case, named Kupenota. The discovery of kuru opened new vistas of human medicine and was pivotal in the subsequent transmission of Creutzfeldt–Jakob disease, as well as the relevance that bovine spongiform encephalopathy had for transmission to humans. The transmission of kuru was one of the greatest contributions to biomedical sciences of the 20th century.

## 1. Introduction

“Kuru”, the first prion disease, was discovered by D. Carleton Gajdusek (Figure 1) [1,2,3,4,5,6,7,8,9,10,11].

It was also the first human prion disease transmitted to chimpanzees and was classified as “a transmissible spongiform encephalopathy” (TSE), or slow unconventional virus disease. It was first reported in two publications by Gajdusek and Vincent Zigas in 1957 (Figure 2) [12,13].

The definition of kuru as both neurodegenerative (not inflammatory) and infectious [14,15] led to subsequent transmission of Creutzfeldt–Jakob disease (CJD) [16] and suggested that kuru represents a novel class of diseases caused by a novel class of pathogens called prions. Kuru won a Nobel prize for Gajdusek in 1976 and, indirectly, as he discovered “prions”, for Stanley B. Prusiner in 1997. Kuru was linked, also indirectly, to a third Nobel Prize for Kurt Wüthrich, who determined the structure of the prion protein [17]. As Gajdusek emphasized for the last time in his life at the Royal Society meeting on kuru in 2007 [18], the research on kuru advanced the ideas of molecular casting, dermatoglyphics, osmium shadowing in electron microscopy, and amyloid-enhancing factors. Kuru research impacted the concepts of nucleation–polymerization and led to an idea of conformational disorders [19,20,21] or prionoids [22].

## 2. Background and Ethnographic Setting

“Kuru”, in the Fore language (Figure 3 and Figure 4), means to tremble from fever or cold [23,24,25,26,27,28,29,30,31,32,33,34,35,36]: “The natives of almost all of the Fore hamlets have stated that it has been present for a ‘long time’, but they soon modify to mean that in recent years it has become an increasingly severe problem and that in the early youth of our oldest informants there was no kuru at all” [15].

The disease was confined to natives of the Fore linguistic group in Papua New Guinea’s Eastern Highlands and neighboring linguistic groups (Auiana, Awa, Usurufa, Kanite, Keiagana, Iate, Kamano, and Gimi). The Fore reports suggested that kuru first appeared at Uwami, a Keiagana village to the northwest, around 1900, and then in the North Fore around 1920. It then traveled down the southeastern border, arriving at Wanitabe in the central South Fore by 1930. From Wanikanto, where it appeared in 1927, it turned toward the northwest, to Miarasa and Paigata. The first cases at Purosa, six miles south of Wanitabe, appeared in the 1930s, and, in some southwestern and southeastern areas, it arrived as late as the 1940s (see map [37]). Zigas and Gajdusek [38] noticed that, when the Fore people at Kasarai moved temporarily to live with the Yar people and settled there for about a decade, kuru cases still occurred. As Lindenbaum observed [39], the Fore descriptions of their encounter with the new disease were rich in detail. It is now thought that kuru first arose in a single individual from a spontaneous change that created a pathogenic, infectious agent in the brain, in the same way that sporadic Creutzfeldt–Jakob disease occurs. The recycling of the infectious agent through the consumption of deceased relatives amplified the agent and the disease in the community, leading to the epidemic [40].

The slow march of kuru was inconsistent with a contemporary genetic hypothesis of Bennett et al. [41,42], but it was consistent with a slow infectious disease. Kuru became endemic in all villages which it entered and became hyperendemic in the South Fore region. When kuru first appeared, the symptoms were thought to resemble the quills of the cassowary, which also reminded them of the swaying fronds of the casuarina tree, so they fed the victims casuarina bark, a homeopathic treatment that gave little relief. Many people also called the disease “negi nagi”, a Fore term meaning silly or foolish person, because the afflicted women laughed immoderately. In those early days, they joked and took sexual levities with the sick women, as they did with those who had temporary mental derangement. When it became apparent that the victims were all dying, they were forced to co-include that sorcerers were at work [37].

## 3. Cannibalism

Endocannibalism (the eating of relatives) was a component of South Fore mourning rituals, in contrast to exocannibalism (the eating of enemies), which was practiced in the north, the difference having consequences for the transmission of the disease. By the 1950s, cannibalism ceased in the North Fore, but was still practiced surreptitiously in the south, where the South Fore said that they continued to hide and eat deceased kin throughout the 1950s [37,43,44,45,46,47,48].

When a body was considered for human consumption, none of it was discarded, except the gall bladder, which was considered too bitter. In the deceased person’s old sugarcane garden, maternal kin dismembered the corpse. They first removed the hands and feet, then cut open the arms and legs to strip the muscles. Opening the chest and belly they avoided the gall bladder. They next severed the head, fracturing the skull to remove the brain. Meat, viscera, and brain were all eaten. Marrow was sucked from cracked bones, and sometimes the pulverized bones were cooked and eaten with green vegetables.

The first Europeans to witness kuru were Australian patrol officers who mentioned the disease in their official reports to the Department of Native Affairs. In 1951, Arthur Carey drew attention to what appeared to be a new disease. Using the term “kuru” for a disease that the Fore said was mainly killing women, Carey described the clinical symptoms and called for a medical diagnosis that could provide medical treatment. John McArthur described the disease in 1953, followed by William Brown in 1954. McArthur’s report confirmed that kuru sorcery was practiced, and noted that he had sent some sorcery offenders to Kainantu (the center of government administration for the Eastern Highlands at that time). In 1955, John Colman sent a typical case of the disease to Kainantu for medical observation, eliciting Vincent Zigas’s provisional diagnosis of acute hysteria in an otherwise healthy woman. Frank Earl, an emergency medical practitioner who accompanied Colman, described the disease in 1955, suggesting that kuru might be a form of encephalitis [49,50]. In the 2000s, Liberski asked Gajdusek when the hypothesis of cannibalism as a vehicle to spread kuru was first envisaged. His response that “even completely drunk would come to the conclusion that a disease endemic among cannibals must be spread through eating corpses”. Gajdusek said this some 50 years after the discovery of kuru. However, the first investigators to propose the hypothesis were Glasse and Lindenbaum [43,48,51]. According to Gajdusek, the hypothesis was taken for granted, but it is also true, that in his Nobel Prize lecture, he said that kuru spread by “conjunctival, nasal, and skin contamination with highly infectious brain tissue”. He was still skeptical about the hypothesis. At that time, William Arens, an anthropologist, said that cannibalism as an accepted social custom did not exist, a view that was finding a receptive public audience, but one that is discredited today [52].

The Fore believed that kuru was the result of sorcery carried out by an envious opponent [37]. A would-be sorcerer, living nearby, could gain a part of the victim’s body—nail clippings, hair, feces, saliva, or partially consumed food such as the discarded skin of sweet potato or even a piece of clothing. These items were enclosed in leaves and bindings made into a “kuru bundle” which was placed in swampy ground. As the bundle disintegrated, the characteristic kuru tremor registered in the victim’s body.

Divination rituals were sometimes enacted to identify a sorcerer. If a suspect approached the corpse and it emitted body fluids (which sometimes happens after death), the guilty person was found. Another divination method was to place hair clippings from a kuru victim into a bamboo cylinder, and a freshly killed possum in another, then calling the name of a suspected sorcerer. If the possum meat remained uncooked, it was thought that the sorcerer’s location was identified. Divination rituals in general identified places of residence, rather than the specific names of individual men.

## 4. Kuru Etiology—The Insight into a Novel Class of Pathogens

Although, on epidemiological grounds, the etiology of kuru was thought to be infectious, patients had no meningoencephalitic signs or symptoms (fever, confusion, convulsions, or coma), no cerebrospinal fluid pleocytosis or elevated protein level, and, on autopsy, no perivascular cuffings or other signs of inflammatory brain pathology. Neither the environmental [53,54,55] nor then available genetic studies [24,56,57,58,59,60,61,62,63,64] provided any clues. Moreover, all attempts to transmit kuru to laboratory rodents or to isolate any microorganism including a virus, using tissue cultures or embryonated hen’s eggs, were unsuccessful. In other wide-ranging investigations, neither exhaustive genetic analyses nor the search for nutritional deficiencies or environmental toxins resulted in a tenable hypothesis [54,55].

On 21 July 1959, Gajdusek, who was at that time in New Guinea, received a letter from the American veterinary pathologist, William Hadlow, at the Rocky Mountain Laboratory in Hamilton, Montana [32,65,66,67,68,69], which stressed the similarities between kuru and scrapie, a slow neurodegenerative disease of sheep and goats known to be endemic in the United Kingdom since the 18th century [70] and experimentally transmitted in 1936 [71,72]. Having seen photographs of kuru plaques at the Wellcome Medical Museum exhibition in London, he enclosed a copy of a letter pointing out the similarity of plaques of kuru and scrapie to the editor of Lancet [65,68].

“I’ve been impressed with the overall resemblance of kuru, and an obscure degenerative disorder of sheep called scrapie… The lesions in the goat seem to be remarkably like those described for kuru… All this suggests to me that an experimental approach similar to that adopted for scrapie might prove to be extremely fruitful in the case of kuru… because I’ve been greatly impressed by the intriguing implication, I’ve submitted a letter to The Lancet”.

A similar observation was made by veterinary neuropathologist Innes [73] during his visit to the Gajdusek laboratory [74] (Gajdusek—telephone conversation, 2008). Hadlow, in his recollection of that seminal observation, pointed out intracellular vacuoles as those neuropathological changes that attracted his attention some 40 years ago [75,76]. Those intracellular vacuoles in scrapie were first described by Besnoit and his colleagues in 1898 [71]. Dr. Gajdusek replied that “as you may have been able to gather from our articles on kuru, we are pursuing the matter of possible infectious etiology extensively—I am, in fact, a virologist by training. However, we have thus far had poor luck with inoculation experiments and the possibility of doing more extensive inoculation works has, until now, been small. We are, however, proceeding accordingly at the present time and frozen and fresh material are being injected into a number of animal hosts during this year’s work on kuru. In your note to Lancet, which I am deeply grateful to you for bringing to my attention, I note that you have probably not seen our extensive pathological description of kuru which includes some features which were little stressed in the report you have quoted”. Gajdusek used Hadlow’s recommendation to keep small laboratory rodents, apes, and monkeys for a longer time. Furthermore, Gajdusek attempted to get fresh brain for inoculation from a kuru case (letter from Gajdusek, dated 6 August 1959).

In 1961, Gajdusek presented a lecture at the 10th Pacific Science Congress in Honolulu entitled “Kuru: an appraisal of five years of investigation”, with a discussion of the still undiscardable possibility of infectious agent, in which he said, “In spite of all the genetic evidence, both the pathological picture and the epidemiological peculiarities of the disease persistently suggest that some yet-overlooked, chronic, slowly progressive, microbial infection may be involved in kuru pathogenesis. Similar suspicion prevails in our current etiological thinking about a number of less exotic and less rare chronic, progressive degenerative diseases of the central nervous system in man. Thus…, amyotrophic lateral sclerosis, Schilder disease, leukoencephalitis, Koshevnikoff’s epilepsy syndrome in the Soviet Union, the Jakob–Creutzfeldt syndromes, acute and chronic cerebellitis, and even many forms of Parkinsonism, especially the Parkinsonism dementia encountered among the Chamorro population in Guam, continue to suggest the possibility that in man there may be infections analogous to the slow infections of the nervous system of animals which were intensively studied by Bjorn Sigurdsson, the Icelandic investigator who formulated the concept of ‘slow virus infections’”. This contention preceded the discovery of kuru transmissibility by more than four years [77]. Parenthetically, many of the diseases mentioned by Gajdusek are now grouped together under the umbrella of “protein conformational disorders” [19,21,78,79]. At the end, in 1965, in a monograph “Slow, Latent, and Temperate Virus Infections” based upon papers presented at the meeting organized in 1964, Gibbs and Gajdusek [80] wrote in an addendum, “although several of the inoculated primates died of acute infection during the period of observation… none developed signs suggestive of chronic neurological disease until the recent onset in two chimpanzees. The first of these, inoculated 20 months previously with a suspension of frozen brain material from a kuru patient, has developed progressive incapacitating cerebellar signs with ataxia and tremor; the second, similarly inoculated with a suspension of brain material from another kuru patient, has developed, 21 months after inoculation, slight wasting lassitude, and some tremor which appeared to be progressive. Whether these syndromes are spontaneous or related to the inoculation remains to be determined”.

## 5. Epidemiology of Kuru—A Strong Support of the Cannibalism Theory

Kuru incidence increased in the 1940s and 1950s [12,14,81,82,83,84] to approach the mortality rate in some villages 35/1000 among a population of some 12,000 Fore people [74,85]. This mortality rate distorted certain populational parameters; in the South Fore, the female–male ratio was 1:1.67 in contrast to the 1:1 ratio in unaffected Kamano people. This ratio increased to 1:2 and even to 1:3 in the South Fore. Gajdusek even calculated the deficit of women in the population to be 1676 persons [33]. The almost total absence of kuru cases in South Fore among children born after 1954 and the rising of age of kuru cases year by year suggested that transmission of kuru to children stopped in the 1950s [86,87,88] when cannibalism ceased to be practiced among the Fore people. Also, brothers with kuru tended to die at the same age which suggested that they were infected at similar age but not at a similar time. The assumption that affected brothers were infected with kuru at the same age led to a calculation of minimal age of exposure for males to be in a range of 1–6 years with a mean incubation period of 3–6 years and the maximum incubation period of 10–14 years [89].

Alpers and Gajdusek wrote a year before the transmission of kuru was published [81]: “The still baffling, unresolved problem of the etiology of kuru in the New Guinea Highlands has caused as to wonder whether or not any or many of the unusual features of its epidemiological pattern and its clinical course may not be changing with time, or even altering drastically under the impact of extensive rapid cultural change, the result of ever increasing inroads of civilization upon the culture of the Fore people”. The comparison of number of deaths from kuru in the period 1961–1963 and 1957–1959 showed a total 23% reduction, while, among the children, a 57% drop, while the kuru mortality rates decreased from 7.64 to 5.58 deaths per thousand. These alterations were not uniform; the North Fore reduction exceeded the South Fore reduction and it is worthy to recall that the South Fore kuru deaths accounted for the 60% of the total. This trend tended to be observed until the disappearance of the kuru epidemic [90].

The proof that kuru was transmitted by cannibalism was provided by Klitzman et al. [91] who studied clusters of kuru patients who participated in kuru cannibalism in the 1940s and 1950s. Three such clusters were found. In one of three, two brothers, Ob and Kasis from Awande village, North Fore developed kuru in 1975, 21 or 27 years after the later or the earlier exposure, respectively. They participated in two cannibalistic feasts.

Of interest, Klitzmann et al. [91] reported that, taking into consideration the fact that Fore women participated in numerous kuru feasts, it is strange that any of them survived into the 1970s. Modern molecular genetics explained this fact in terms of the codon 129 polymorphism of the *PRNP* gene. In the younger patients, homozygotes 129^Met Met^ predominate; the latter finding is reminiscent of that of variant CJD (vCJD) [92] and suggests the increased susceptibility of 129^Met Met^ individuals, with a shorter incubation period than other *PRNP* codon 129 genotypes.

## 6. Transmission Experiments

The transmission of kuru to chimpanzee won a Nobel Prize for Gajdusek in 1976 [34,35,77,93,94,95,96,97,98,99,100]. The list of non-primates to which kuru was transmitted over the years is given in Table 1.

The list of non-human primates (Table 2) include rhesus monkeys [101], marmosets [102], gibbon and sooty mangabey monkeys [103]. The detailed description of experimental kuru in 41 chimpanzees was published in 1973 [2]. The incubation period varied from 11 to 39 months (the average was 23 months for the first passage, 12 months for the second passage, 13 months for the third passage, and the same for the fourth passage; the clinical course was divided into three stages.

Early stage (I)

(a)Prodromal period defined by earliest alterations in behavior; animals became inactive, often “extremely dirty” and submissive. “Vicious and aggressive animals became passive and withdrew from competition with their normal cagemates, allowing smaller chimpanzees to tease and take food from them… periods of sullen apathy were often interrupted by outbursts of furious screaming”.(b)Period of minimal disabilities characterized by minor motor dysfunction; animals did not leave cages, “to run or to climb”, they were slow and fell with forced movements; the movements were “like… in slow-motion cinema”.

Intermediate stage (II)

The beginning of this stage was defined by difficulties when a chimp raises from a supine position; gait is ataxic but sitting is still possible. The gait of chimpanzees is quadrupedal, i.e., “knuckle walking”, where animals put hands on the ground not with palms but with knuckles. Truncal titubation, typical for human kuru, is seen as of stage II. Muscle tone is increased and flexion contractures develops. Severe tremor is observed along with choreiform movements, vision difficulties, lateral nystagmus, and intermittent left strabismus; however, but Babinski signs were only occasionally observed.

At the late stage (III), animals cannot raise by themselves from a supine position or are unable to sit, and decubitus ulcers develop. They eat inedible objects. A severe startled response of flexion of all extremities accompanied by violent coarse trembling of all limbs develops.

Kuru neuropathology (Figure 5a–c) in chimpanzees was described by late Elisabeth Beck and Daniel [104,105,106,107,108,109,110]. The neuropathological picture was practically identical to that of natural kuru except for the absence of amyloid plaques. In the cerebral cortex, the spongiform change and intraneuronal vacuoles, similar to those encountered in scrapie, were the most prominent, accompanied by a severe astrocytic gliosis. Binucleated neurons were prominent; the same type of neuronal lesions was also seen in the spider monkey [111].

## 7. Clinical Manifestations

“I was still very young when I saw [kuru] and even after we treated it there was no help. Everyone was falling apart. [Kuru victims] were aware there was no cure and that they would die. It wasn’t just one person that this sickness came to—there were about three in a house line and then after they died there would be another three. It was… ongoing… there were many deaths. Once a [person]…was affected by kuru [their] family would think that the clan had poisoned [them] and they would start… shooting at each other and that made it worse. It was chaos!” (Taurubi) [112].

Kuru is an always fatal cerebellar ataxia accompanied by tremor, choreiform, and athetoid movements (Figure 6, Figure 7, Figure 8, Figure 9, Figure 10, Figure 11 and Figure 12) [12,25,29,32,70,112,113,114,115,116,117,118,119,120,121,122].

In contrast to the neuropathological picture, the neurological picture is highly uniform. The progressive dementia typical for most subtypes of sporadic CJD is barely noticeable, and, if it is present, then it is only late. In contrast, kuru patients often displayed emotional changes, including inappropriate euphoria and compulsive laughter (the journalistic “laughing death” or “laughing disease”), or apprehension and depression. Kuru is divided into three clinical stages: ambulant, sedentary, and terminal (the Pidgin expressions, wokabaut yet, i.e., “is still walking”; sindaun pinis, i.e., “is able only to sit”; and slip pinis, i.e., “is unable to sit up”) [113,123]. The duration of kuru, as measured from the onset of prodromal signs and symptoms until death was about 12 months (3–23 months) [113,123].

There is an ill-defined prodromal period (kuru laik i-kamp nau—i.e., “kuru is about to begin”) characterized by headache and limb pains, often in the joints; frequently, knees and ankles came first, followed by elbows and wrists; sometimes, interphalangeal joints were initially affected, along with abdominal pains and loss of weight. The prodromal period lasted several months. Fever and typical signs of infectious disease were not observed, but the patient’s general feeling resembled that of acute respiratory infection. Some patients used to say that they awaited cough; when cough did not come, they awaited approaching kuru.

The prodromal period is followed by the “ambulant stage”, which ends when the patient is unable to move without support of a “family”, which also initiated the search for a sorcerer. This ambulant stage is characterized by the beginning of subtle signs of gait unsteadiness that are noticed first by the patient. Those symptoms, in about a month, advanced to marked astasia and ataxia and incoordination of the muscles in the trunk and lower limbs. As patients knew that kuru means death in about a year, they became withdrawn and quiet. A delicate “shivering” tremor, starting in the trunk, amplified by cold and associated with a “goose flesh”, is followed by titubation and other types of abnormal movements, such as clawing of the toes and curling of the feet. Plantar reflex is always flexor while clonus, especially ankle clonus, is typical, but observed only temporarily.

In the early stages, ataxia could be demonstrated only when the patient stood on one leg; the Romberg sign was almost always negative (two of 34 kuru cases in Alpers’ series [123]); however, with the progression of disease, ataxia became marked and the Romberg sign became positive. Ataxia in the upper limbs followed that in the lower limbs; dysmetria was often the initial sign. Dysarthria appeared early. Involuntary spontaneous and action jerks are a typical sign of kuru. According to Alpers “[it is] difficult to describe and analyze. It appears to include the following components: a shivering component, an ataxic component, and, in the latter stages and certain cases only, the extrapyramidal component. A fine shivering-like tremor may be present from the onset of disease… it is potentiated by cold and thus may not be found in the heat of the day; a sudden drop in temperature not sufficient to make others shiver will induce it in kuru patients. As ataxia increases a more obvious ataxic component is added and the shaking movements become wilder and more grotesque”. The major component in kuru is the intention (cerebellar tremor) and jerky movements or jerks (originally described as tremor or shivering) occurring at rest, when maintaining posture, triggered by movement or other stimuli (lights, noise, or touch). It subsides; “It often seemed to be triggered by minor movement, an adjustment of posture, stretching out the arm in greeting, or even a sudden turning of the eyes”. Patients learn how to control tremor. A child trembling violently, described by Zigas and Gajdusek [13,38], found that he may almost completely abolish the tremor when curled into a flexed, fetal position in his mother’s lap. A horizontal convergent strabismus is a typical sign, especially in younger patients; nystagmus was common but the papillary responses were preserved. Facial hemispasm and supranuclear facial palsies of different kinds were also observed.

The second “sedentary” stage begins when the patient is unable to walk without constant support and ends when he or she is unable to sit without it. “The gait was, by definition non-existent. However, if a patient was ‘walked’ between two assistants a caricature of walking was produced, with marked truncal instability, weakness at hips and knees and heavy leaning on one or other assistant for support; but steps could be taken, and were characterized by jerky flinging, at times decomposed movements, which led to a high-steppage, stamping gait”. Postural instability, severe ataxia, tremor, and dysarthria progress endlessly through this stage. Deep reflexes may be increased, but the plantar reflex is still flexor. Jerky ocular movements: were characteristic. Opsoclonus, a rapid horizontal ocular agitation was also occasionally noticed. Zigas and Gajdusek [13,38] reported peculiar, jerking, clonic movements of the eyelids and eyebrows in patients confined to the dark indoors of huts and then transported outdoors to the light. Two cases of 34 showed signs of dystonia.

In the third stage, the patient is bedridden and incontinent, with dysphasia and primitive reflexes, and eventually succumbs in a state of advanced starvation. “The patient at the beginning of the third stage usually spent the day supported in the arms of a close relative”. Extraocular movements were jerky or slow and rigid. Deep reflexes were exaggerated, but Babinski signs were never observed. Generalized muscle wasting became seen and fasciculation, spontaneous or evoked by tapping, was observed. Some symptoms of dementia were also noticed, but in terminal stages, patients many understood the Fore language and tried to answer appropriately. A strong grasp reflex occurred, as well as fixed dystonic postures, athetosis, and chorea. In one case “almost constant small involuntary movements, involving mouth, face, neck, and hands” were seen.

Terminally, “the patient lies moribund inside her hut surrounded by a constant group of attending relatives… She barely moves and is weak and wasted. Her pressure sores may have spread widely to become huge rotting ulcers which attract a swarm of flies. She is unable to speak. The jaws are clenched and have to be forced open in order to put food or fluid in… Despite her mute and immobile state, she can make clear signs of recognition with her eyes and may even attempt to smile”.

It is worth mentioning incredible support given by Fore to dying kinsmen. “The family members live with the dying patient, siblings sleep closely huddled to their brother or sister in decubitus, parents sleep with their Kuru-incapacitated child cuddled to them and a husband will patiently lie beside his terminal, uncommunicative, incontinent, foul smelling wife” [38].

## 8. Neuropathology

The first examination of kuru neuropathology (12 cases) was published by Klatzo et al. in 1959 [124,125].

Neurons were shrunken and hyperchromatic or pale, with dispersion of Nissl substance with intracytoplasmic vacuoles similar to those already described in scrapie. In the striatum, some neurons, as well as Purkinje cells in the cerebellum, were vacuolated to such a degree that they looked “moth-eaten” (Figure 13). Spongiform change was seen (Figure 14).

Neuronophagia was observed. A few binucleated neurons were visible, and torpedo formation was noticed in the Purkinje cell layer, along with empty baskets that marked the presence of absent Purkinje cells. In the medulla, neurons of the vestibular nuclei and the lateral cuneatus were frequently affected; the spinal nucleus of the trigeminal nerve and nuclei of VIth, VIIth, and motor nucleus of the VIth cranial nerves were affected less frequently, while nuclei of the XIIth cranial nerve, the dorsal nucleus of Xth cranial nerve, and nucleus ambiguous were relatively spared. In the cerebral cortex, the deeper layers were affected more than the superficial layers, and neurons in the hippocampal formation were normal. In the cerebellum, the paleocerebellar structure (vermis and flocculo-nodular lobe) was most severely affected, and spinal cord pathology was most severe in the corticospinal and spinocerebellar tracts. Astroglial (Figure 15a,b) and microglial proliferation was widespread; the latter formed rosettes and appeared as rod or amoeboid types or as macrophages (gitter cells). Myelin degradation was observed in 10 of 12 cases. Interestingly, the significance of vacuolar changes was not appreciated by Klatzo et al. [124,125], but “small spongy spaces” were noted in seven of 13 cases studied by Beck and Daniel [106,107,108,109,110].

The most striking neuropathologic feature of kuru was the presence of numerous amyloid plaques found in six of 12 cases studied by Klatzo et al. [124,125], and in “about three-quarters” of the 13 cases of Beck and Daniel [106,107,108,109,110]; they became known as “kuru plaques” (Figure 16a–f) [126,127,128,129,130,131,132].

Plaques had a diameter of 20–60 μm, were of round or oval in shape, and consisted of a darker core with subtle radiating periphery enveloped by a pale “halo”. Kuru plaques were most common in the granular cell layer of the cerebellum, basal ganglia, thalamus, and cerebral cortex in that order of frequency. Kuru plaques are metachromatic and stain with PAS, Alcian blue, and Congo-red, and are weakly argentophilic when impregnated according to Belschowsky or von Braunmühl techniques. Of historical interest, another unique disease reported by Seitelberger [133] as “a peculiar hereditary disease of the central nervous system in a family from lower Austria” (germ. Eigenartige familiar-hereditare Krankenheit des Zentralnervensystems in einer niederoosterreichen Sippe) was mentioned by Neumann et al. [134], who was, thus, the first person to suggest a connection between kuru and Gerstmann-Straüssler-Scheinker disease. Indeed, the latter was transmitted to non-human primates in 1981 [135].

Renewed interest in kuru pathology was raised by the appearance of a novel form of CJD, variant CJD, characterized by numerous amyloid plaques, including “florid” or “daisy” plaques—a kuru plaque surrounded by a rim of spongiform vacuoles [132]. To this end, a few papers re-evaluating historic material were published [136]. We [137] studied, using PrP-immunohistochemistry, the case of a young male kuru victim of the name Kupenota from the South Fore region whose brain tissue transmitted disease to chimpanzees, and McLean et al. [138] examined 11 cases of kuru. Both current papers described typical spongiform change (Figure 14) present in deep layers (III–V) of the cingulate, occipital, enthorrinal, and insular cortices, and in the subiculum. Spongiform change was also seen in the putamen and caudate, and some putaminal neurons contained intraneuronal vacuoles. Spongiform change was prominent in the molecular layer of the cerebellum, in peraqueductal gray matter, basal pontis, central tegmental area, and inferior olivary nucleus. The spinal cord showed only minimal spongiform change.

There are no ultrastructural observations on kuru in humans using fresh material except the paper by Peat and Field [139] who stressed the presence of the “intracytoplasmic dense barred structures” and otherwise normal structures of Purkinje cells [140]; Field et al. [139] described the typical ultrastructure of the kuru plaques and “herring-bone” structures, again with either the normal structure of the neuron or Hirano bodies [141]. In kuru in chimpanzees, Lampert et al. [142] and Beck et al. [111] found severe confluent spongiform change corresponded to typical membrane-bound vacuoles. Neurites showed dystrophic changes. Our studies on formalin-fixed paraffin-embedded kuru specimens reversed to electron microscopy revealed typical plaques composed of amyloid fibrils (Figure 16).

Immunohistochemical studies revealed that misfolded PrP^Sc^ was present not only as kuru plaques but also in synaptic and perineuronal sites (Figure 16a–f) [129,137], and, in the spinal cord, the substantia gelatinosa was particularly affected, as in iatrogenic CJD cases following peripheral inoculation [143]. Brandner et al. [144] studied one of the very last cases of kuru and basically confirmed the findings of Hainfellner et al. [137]. The latter case was neuropathologically compared with known subtypes of CJD and it seems the most similar to type 3 129 MV type of CJD of the Collinge et al. [145] classification or type 2 CJD of the Parchi et al. [146] classification [147]. Of note, immunocytochemistry with 12F10 antibodies revealed a stronger signal than that using 3F4 anti-PrP antibodies [146].

## 9. Genetics and Molecular Biology of Kuru

Even after 40 years, the summary of the genetics of kuru written by Michael P. Alpers [81] is still valid: “it was recognized that a strong familial association of disease does not necessarily prove that the cause is genetic (of note, Gajdusek used to stress that the French language is also genetic, i.e., familial). Furthermore, it was hard to see how a disease so prevalent and at the same time so lethal could have become established in the population by purely genetic means, unless there was some immense associated heterozygote advantage”. At the beginning, it was discovered that two kuru cases were 129^Met Met^ [148]. Further studies found that individuals of 129^Val Val^ and 129^Met Val^ genotype were susceptible to kuru, but those of 129^Met Met^ genotype were overrepresented in the younger age group, while those of 129^Val Val^ 129^Met Val^ were overrepresented in much older age group [47,92,149,150,151,152]. In contrast, those people who survived the epidemic were characterized by almost the total absence of 129^Met Met^ homozygotes. The more recent cases studies by Lantos et al. [153], McLean et al. [138], and us [137] were all 129^Met Met^ homozygotes. Recent genome-wide studies confirmed a strong association of kuru with an SNP localized within the codon 129, but also with two other SNPs localized within genes *RARB* (the gene encoding retinoic acid receptor beta) and *STMN2* (the gene encoding SCG10) [149].

The practice of endocannibalism underlying the kuru epidemic created a selective pressure on the prion protein genotype [154,155]. As in CJD, homozygosity at codon 129 (129^Met Met^ or 129^Val Val^) is overrepresented in kuru [48,92,149,150,151,152]. Furthermore, Mead et al. [150,151] found that, among Fore women over fifty years of age, there is a remarkable overrepresentation of heterozygosity (129^Met Val^) at codon 129, which is consistent with the interpretation that 129^Met Val^ makes an individual resistant to prions and that such a resistance was selected by cannibalistic rites. Another protective polymorphism G127V located in a highly conserved region of PrP was discovered by the Collinge’s group [156,157]. This 127^Val^ was not found in any of kuru patients. Because of this 129^Met Val^ heterozygote advantage, it was suggested that the heterozygous genotype at codon 129 was sustained by a widespread ancient practice of human cannibalism [158]. Furthermore, there is a hypothesis that the extinction of Neanderthals co-existing with *Homo sapiens* some 45,000 to 30,000 years ago is connected to the appearance of “kuru-like” epidemics spread by cannibalism [159,160]. Collinge et al. [119] suggested that the survival advantage of the *PRNP* 129^Met Met^ heterozygotes provides a basis for a selection pressure not only in Fore, but also in those human populations that practiced cannibalism. Of note, this was preconceived by Alpers and Gajdusek in 1965: “In order to explain the combination of high incidence and high lethality, which at first glance might seem to entirely rule out a genetic cause unless there was an immense heterozygote advantage, we postulated that environmental change, of relatively recent origin, has given a lethal expression to a previously benign gene mutation established in the Fore population as a genetic polymorphism” [81].

The molecular strain typing of kuru cases was performed by the Collinge’s group [161,162]. This typing is based on the electrophoretic mobility of de-, mono-, and diglycosylated bands of PrP^Sc^ following digestion with proteinase K [145]. The four major types of PrP^Sc^ were found. The human PrP^Sc^ types 1 and 4 occur only in individuals of the codon 129^Met Met^ of the *PRNP* gene; type 3 is seen in individuals with at least one Val at this codon, and type 2 occurs in all codon 129 variants. There is another classification based on only two PrP^Sc^ types [146], and the agreement between supporters of either classification is yet to be achieved. The kuru specimens revealed type 2 (129^Met Met^) or 3 (129^Val Val^) PrP^Sc^ patterns and the glycoform ratio was similar to that of sporadic CJD, but not typical for vCJD [163,164,165]. In primates inoculated with kuru and sCJD VV2 and sCJD MV 2K, the “b” pattern of pathology i.e., coarser vacuoles situated in the subcortical structures and in deeper layers of the cerebral cortex, was observed. PrP^Sc^ consisted of doublets of 20 kDa and 21 kDa. The latter notion is supported by the fact of a similar transmission rate of kuru to transgenic mice lacking the mouse *PrP* gene but expressing the human *PrP* 129^Val Val^ gene [161,162]. In contrast, kuru was reported as not transmissible to normal wild-type mice, but it was later shown that it transmits to CD-1 mice with unique clinical and neuropathological patterns in infected animals [166]. Of interest, the robust presence of PrP^Sc^ in follicular dendritic cells in the spleen suggests a possibility of the spreading of the kuru agent via the bloodstream. The presence of PrP^Sc^ in follicular dendritic cells is also true for other TSEs. Collectively, those data suggest that kuru is unique and different from either sporadic CJD or variant CJD.

## 10. Final Remarks

Kuru, an extinct exotic disease of a cannibalistic tribe in a remote Papua New Guinea, still impacts on many aspects of neurodegeneration research. It is, thus, rather curious that, in the newest Handbook of Clinical Neurology [167], there is no kuru chapter, with just a small note on history. Firstly, it showed that a human neurodegenerative disease can result from an infection with an infectious agent, later called a “slow virus” [99]. This discovery opened vistas into the new class of human diseases including Creutzfeldt–Jakob disease, Gerstmann–Sträussler–Scheinker disease, and, recently, fatal familial insomnia. Parenthetically, CJD was pointed out as a possible analog of kuru based of non-specific neuropathological findings, but Gerstmann–Sträussler–Scheinker disease was identified as linked because of the presence of numerous amyloid plaques not unlike kuru plaques. The kuru plaque became a link to Alzheimer disease and, as Gajdusek suggested [20], all amyloidoses share a common pathogenetic mechanism—processing of a normal protein into an amyloid deposit. Indeed, it was recently shown that Aβ amyloid and α-synuclein in MSA spread as prions [168,169,170]. This event underlies all “conformational disorders”, including pathogenetically novel classes of neurodegenerations like α-synucleinopathies, tauopathies, and expanded triplet disorders.

## Figures and Tables

**Figure 1 viruses-11-00232-f001:**
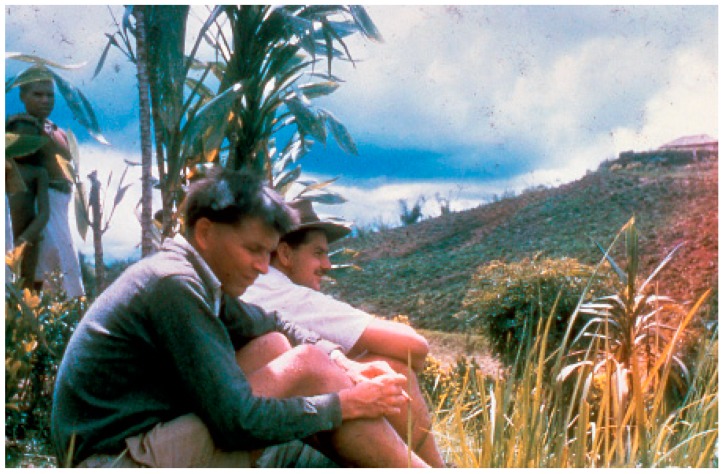
Dr. D. Carleton Gajdusek (lhr-57-ng-332). Courtesy of D. Carleton Gajdusek.

**Figure 2 viruses-11-00232-f002:**
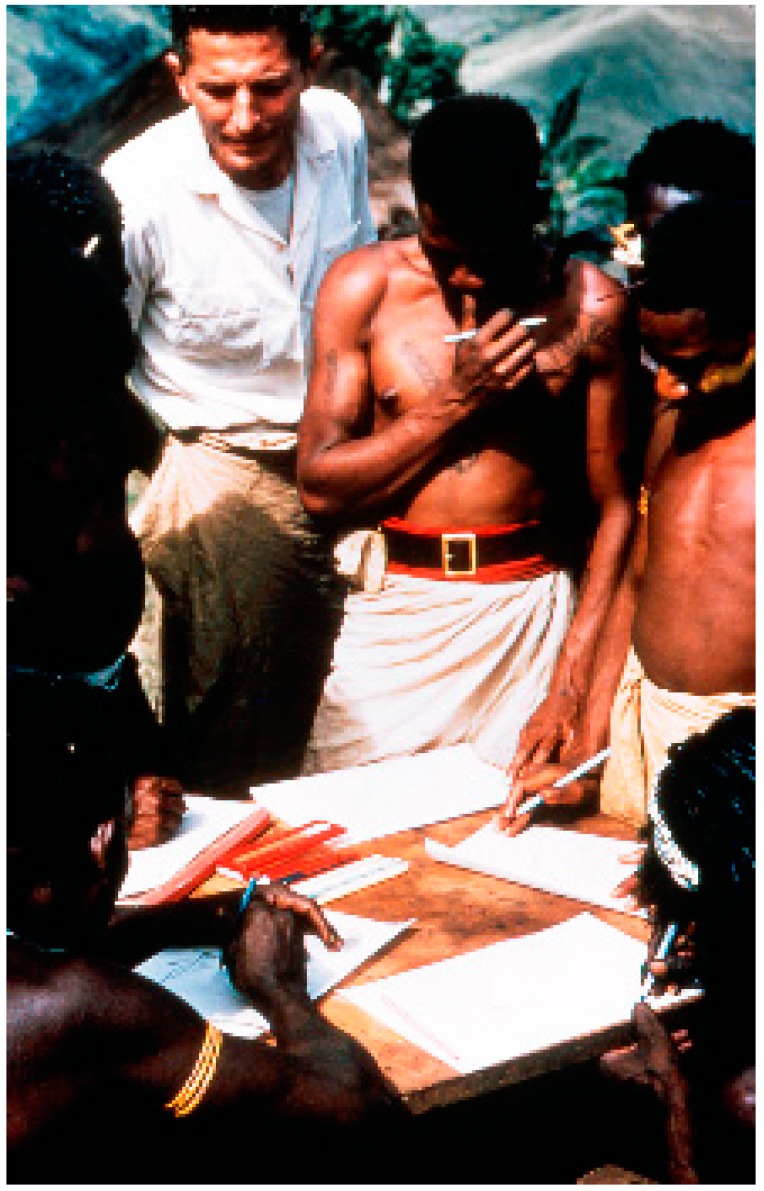
Dr. Vin Zigas (dcg-57-ng-336). Courtesy of D. Carleton Gajdusek.

**Figure 3 viruses-11-00232-f003:**
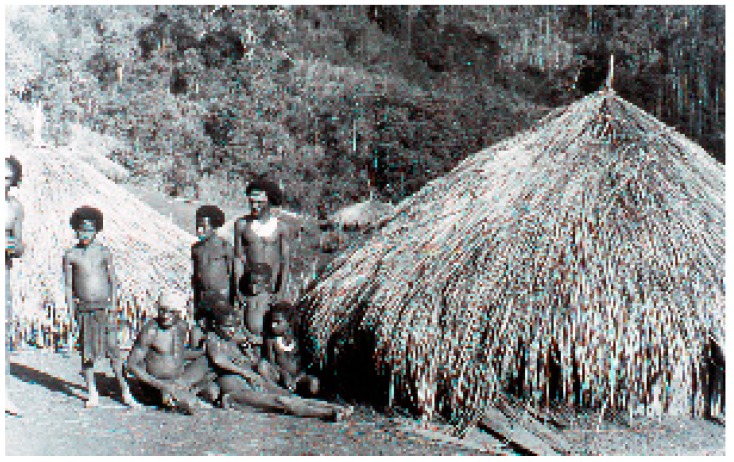
A general view of a Fore hamlet. A kuru victim is sitting in the first row between two supporters (dcg-ng-bw-12). Courtesy of D. Carleton Gajdusek.

**Figure 4 viruses-11-00232-f004:**
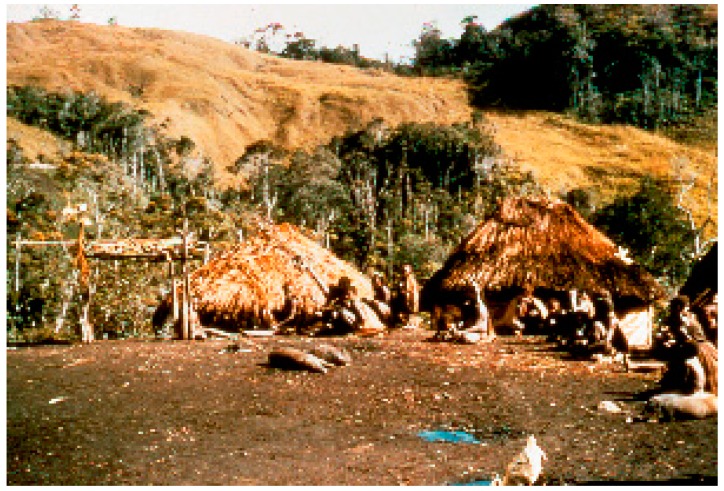
A general view of a Fore hamlet (dcg-57-ng-186). Courtesy of D. Carleton Gajdusek.

**Figure 5 viruses-11-00232-f005:**
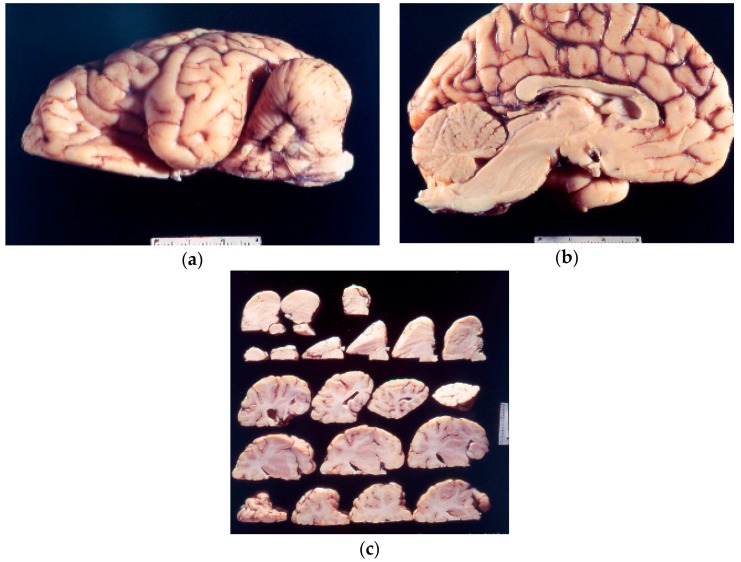
(**a**–**c**) Macroscopic picture of the chimp brain (67-10825-2-1261214; 67-10825-5-1261214; 67-10825-5-126121-24). Courtesy of D. Carleton Gajdusek.

**Figure 6 viruses-11-00232-f006:**
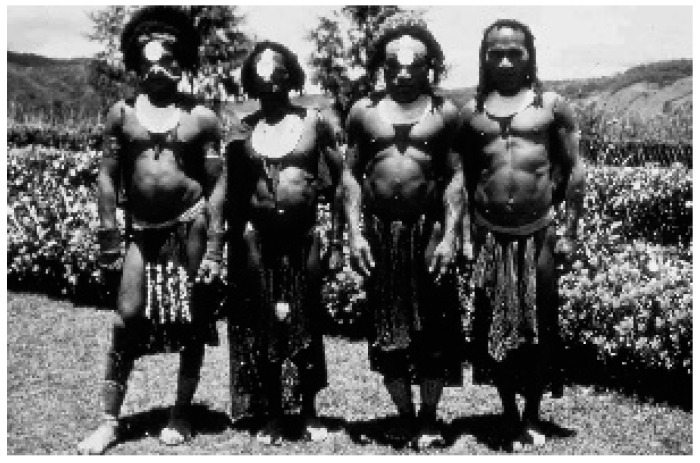
A group of several Fore men (dcg-57-ng-118). Courtesy of D. Carleton Gajdusek.

**Figure 7 viruses-11-00232-f007:**
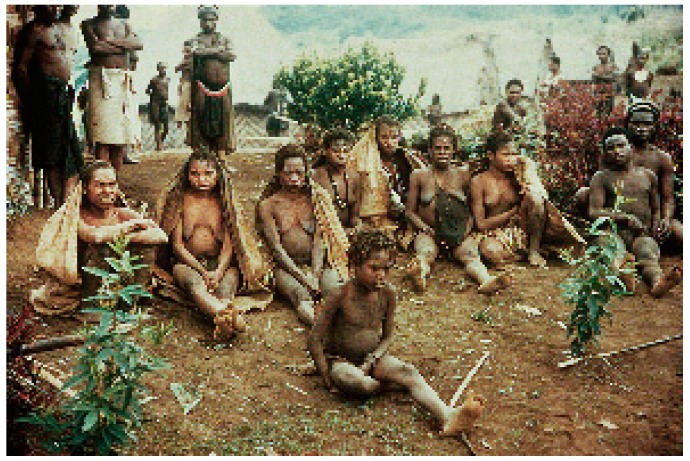
A group of kuru victims. Courtesy of D. Carleton Gajdusek.

**Figure 8 viruses-11-00232-f008:**
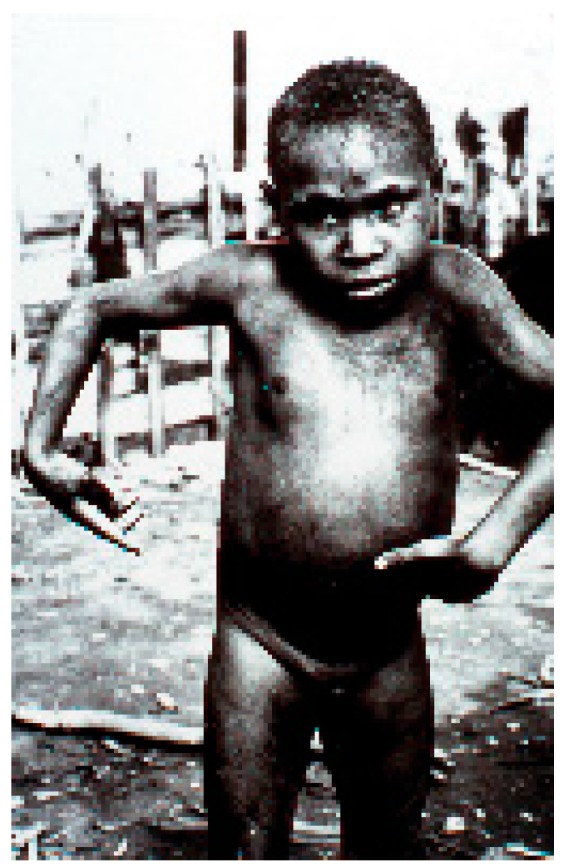
A boy afflicted with kuru showing atethoid movements (dcg-ng-bw-5). Courtesy of D. Carleton Gajdusek.

**Figure 9 viruses-11-00232-f009:**
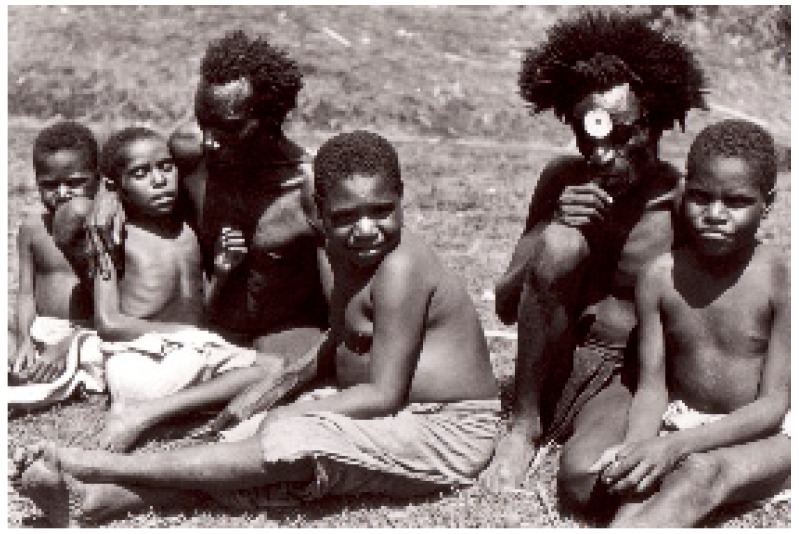
A group of children with kuru (dcg-bw-9). Courtesy of D. Carleton Gajdusek.

**Figure 10 viruses-11-00232-f010:**
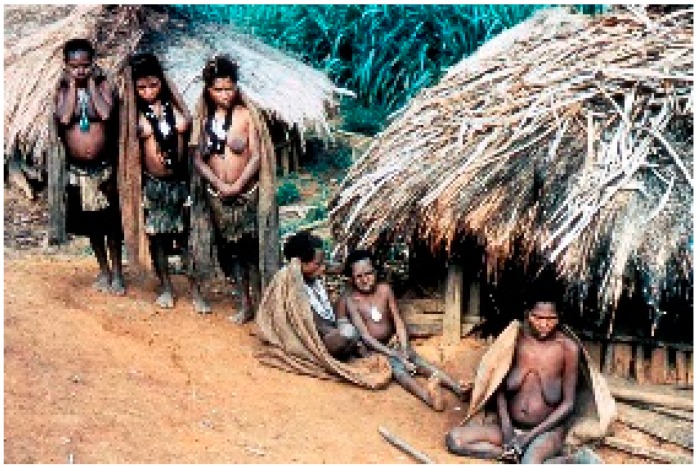
A group of women afflicted with kuru (dcg-57-ng-573). Courtesy of D. Carleton Gajdusek.

**Figure 11 viruses-11-00232-f011:**
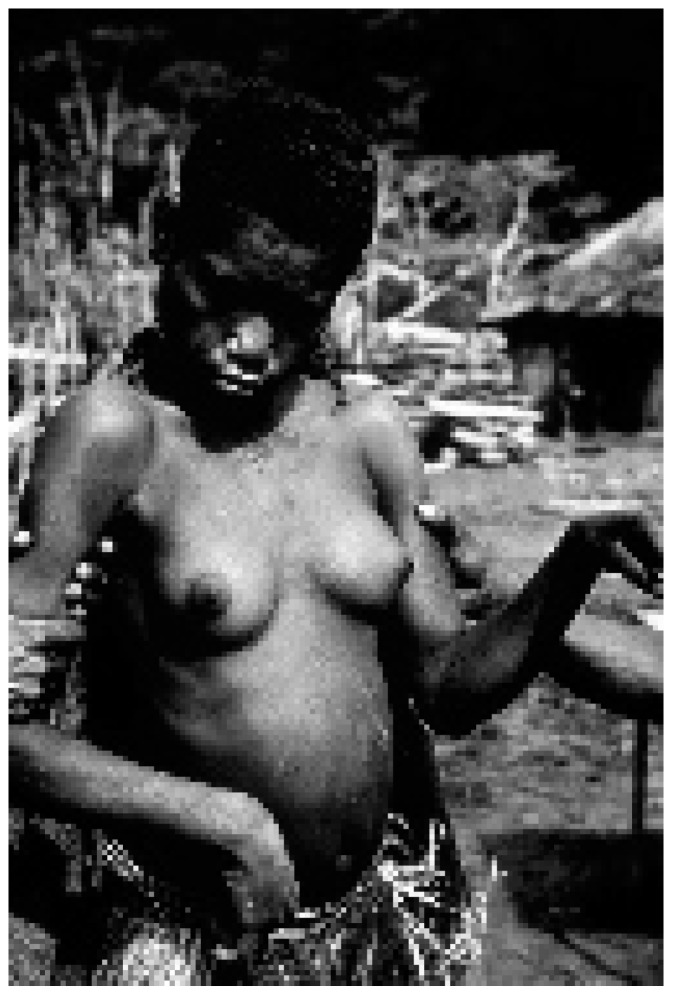
A woman affected with kuru (dcg-ng-57-346A). Courtesy of D. Carleton Gajdusek.

**Figure 12 viruses-11-00232-f012:**
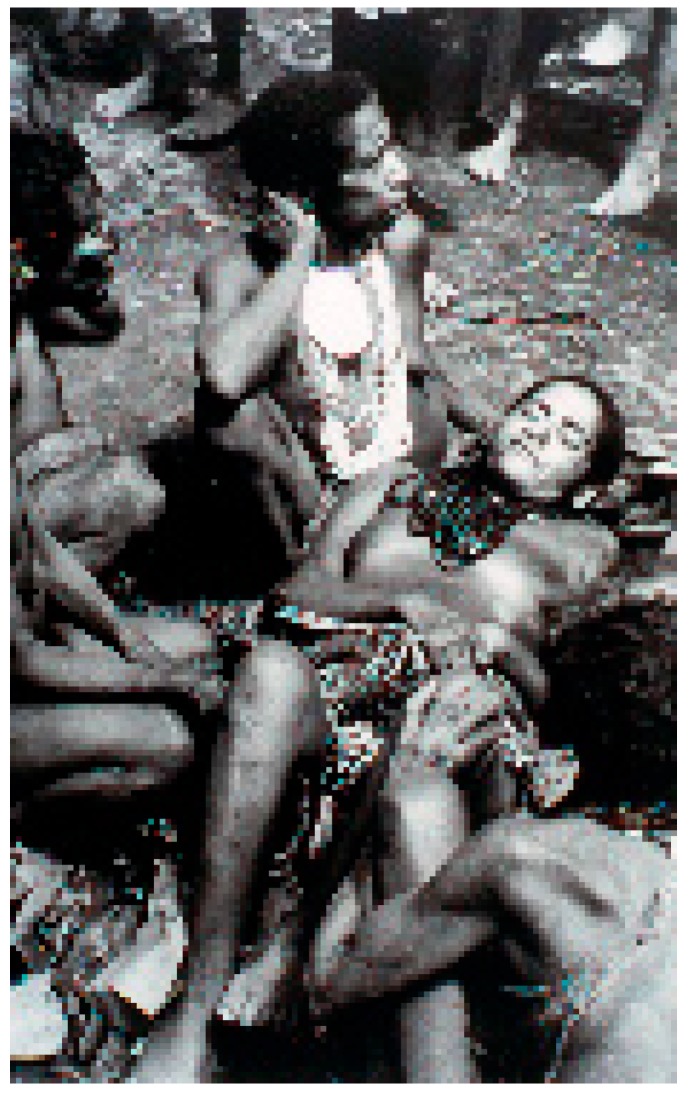
A dead woman with kuru (dcg-ng-bw-11). Courtesy of D. Carleton Gajdusek.

**Figure 13 viruses-11-00232-f013:**
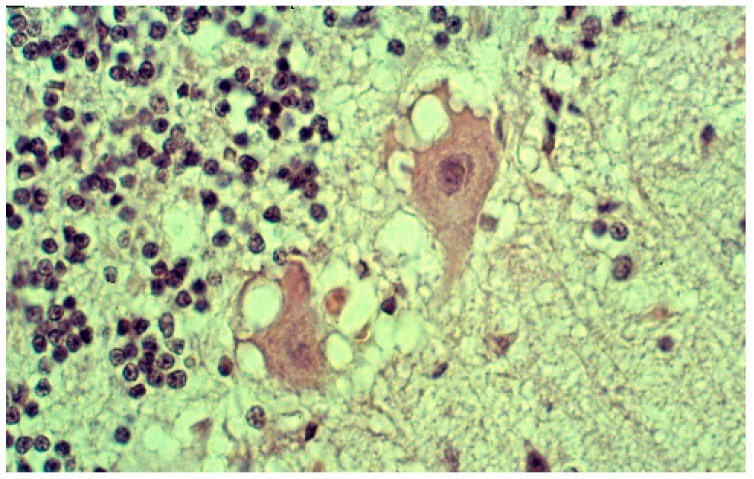
A vacuolated “moth-eaten” neuron. Courtesy of D. Carleton Gajdusek.

**Figure 14 viruses-11-00232-f014:**
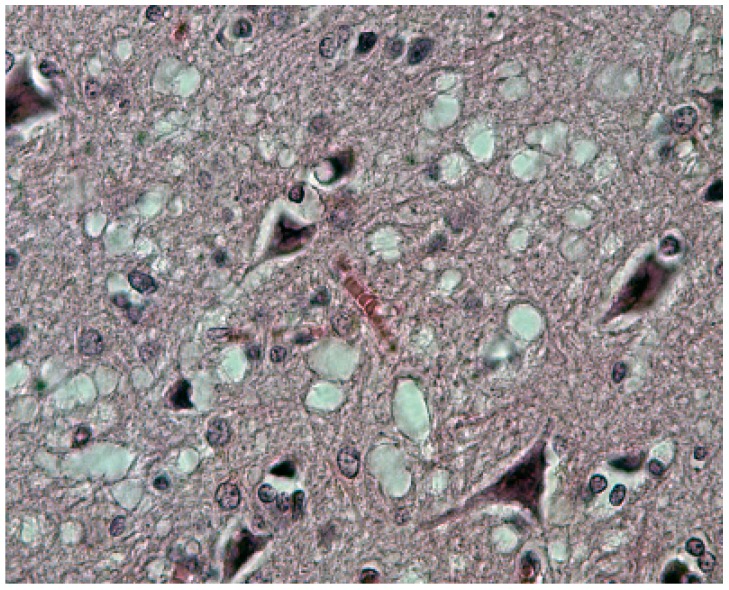
Typical spongiform change (41-93-2a-40-3). Courtesy of D. Carleton Gajdusek.

**Figure 15 viruses-11-00232-f015:**
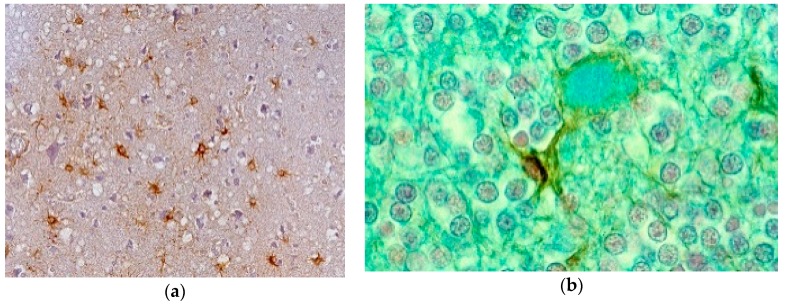
(**a**) Proliferation of GFAP-positive astrocytes against the background of spongiform change; (**b**) amyloid plaque stained with Alcian blue surrounded by an astrocyte. Courtesy of D. Carleton Gajdusek.

**Figure 16 viruses-11-00232-f016:**
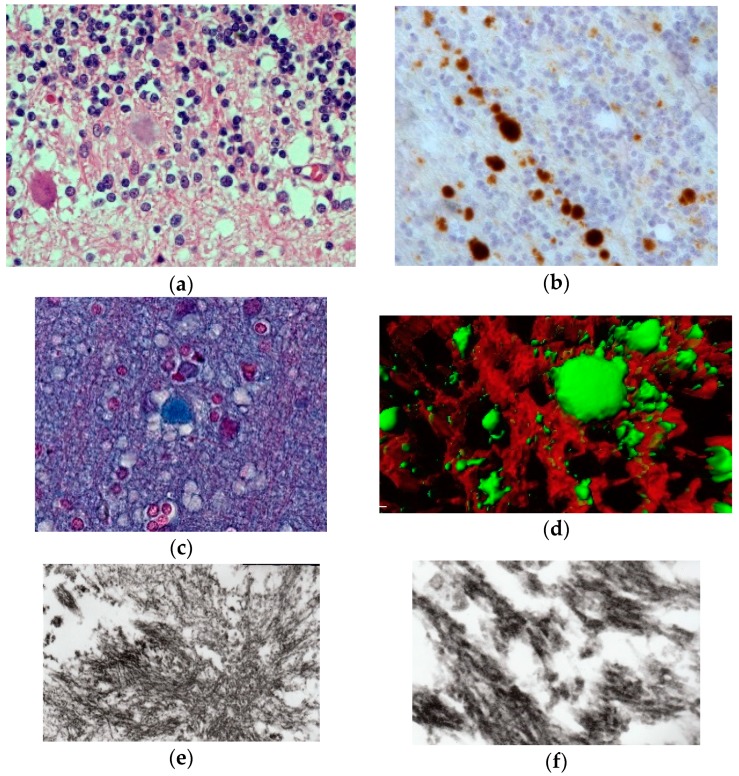
(**a**) Typical kuru plaque stained silver. Courtesy of D. Carleton Gajdusek; (**b**) a linear row of kuru plaques probably attached to a neurite; (**c**) a kuru plaque stained with Alzian blue (41-93-2a-alcian-20-17-alcian). Courtesy of D. Carleton Gajdusek; (**d**) A kuru plaque and PrPSc deposits surrounded by glial cells in the human cerebellum. Confocal laser microscopy, prion protein (clone 12F10, Alexa Fluor 488)—green, GFAP (polyclonal rabbit, AlexaFluor 546)—red, magnification 600×, digital zoom 2.1×. Three-dimensional reconstruction with surface rendering. (**e**) Electron microscopic image of the kuru plaque, material reversed from paraffin, 13,000×; (**f**); higher magnification to show bundles of kuru plaque-fibrils, 50,000×.

**Table 1 viruses-11-00232-t001:** Non-primate host range for kuru transmission; the incubation period is given in months.

Species	Incubation Period (Months)
Goat (*Capra hircus*)	(104+)
Guinea pig (*Cavia porcellus*)	(27)
Opossum (*Didelphis marsupialis*)	(22+)
Domestic cat (*Felis domesticus*)	(59)
Gerbil (*meriones unguiculatus*)	(24+)
Hamster (*Mesocricetus auratus*)	(28)
Mouse (*Mus musculus*)	22.5
Ferret (*Mustela putorius*)	18–70.5
Mink (*Mustela vision*)	45
Sheep (*Ovis aries*)	(63+)

^1^ Number in parenthesis—number of months elapsed since the inoculation, during which the animals remained asymptomatic.

**Table 2 viruses-11-00232-t002:** A host range of the primates susceptible for kuru.

Species	Incubation Period (Months)
**Apes**	
Chimpanzee (*Pan troglodytes*)	10–82
Gibbon (*Hylobates lar*)	(+10)
**New World Monkeys**	
Capuchin (*Cebus albifrons*)	10–92
Capuchin (*Cebus paella*)	11–71
Spider (*Ateles geofffroyi*)	10–85.5
Moramoset (*Saguinus* sp.)	1–176
Wolly (*Lagothrix lagotricha*)	33
**Old World Monkeys**	
African Green (*cercopithecus aethiops*)	18
Baboon (*papia anubis*)	(130)
Bonnet (*macaca radiate*)	19–27
Bushbaby (*Galago senegalensis*)	(120)
Cynomolgus macaque (*Macaca fascicularis*)	16
Patas (*Erythrocebus patas patas*)	(136)
Pigtailed macaque (*Macaca nemestrina*)	70
Rhesus (*Macaca mulatto*)	15–102
Sooty mangabey (*Cerocebus atys*)	(+2)
Talapoin (*Cecopithecus talapin*)	(1+)

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
