# Peer review of "Kuru, the First Human Prion Disease†"

_viruses, 2019, doi:10.3390/v11030232_

Round 1
Reviewer 1 Report
This is a very interesting review which provides great insights into the history of prion diseases, with unique photographs and figures to illustrate Kuru. There is only one request to revise the statement in line 39, implicating that Prusiner won the Nobel prize for work on Kuru which is not entirely correct as Prusiner used scrapie as a model for discovery of prions and isolating the infectious agent.
Author Response
Point 1: This is a very interesting review which provides great insights into the history of prion diseases, with unique photographs and figures to illustrate Kuru. There is only one request to revise the statement in line 39, implicating that Prusiner won the Nobel prize for work on Kuru which is not entirely correct as Prusiner used scrapie as a model for discovery of prions and isolating the infectious agent.
Response 1: Done
Reviewer 2 Report
This is an exceptionally valuable review on Kuru by highly knowledgeable investigators who have actively participated to the clinical and neuropathological study of the disease. Shirley Lindenbaum was one of the pioneer physicians who studied Kuru in Papua New Guinea, and Pawel Liberski had first hand insights into historical data about the disease from D.C. Gajdusek himself.
This review describes epidemiology, disease symptoms, and neuropathology of Kuru. It also describes the history of the pioneers, and information on how disease was managed by the Fore people themselves. The paper is rich in information, provides vibrant descriptions, and will remain a reference in the field. I only have a few minor editorial comments.
1. I suggest explaining the meaning of the numbers present in the figure legends.
2. Fig. 5: please provide legends to b) and c).
3. Lane 109: the word “was” is missing.
4. Lines 114-116: please clarify this sentence.
5. Table 1: “e” of “mouse” is missing.
6. Figure 13: please add “Striatal (?) neurons in the brain of…”
7. Lanes 506-507: The presence of PrPsc in follicular dendritic cells is also true for other TSEs.
Author Response
1. I suggest explaining the meaning of the numbers present in the figure legends.
Response 1: Done
2. Fig. 5: please provide legends to b) and c).
Response 1: Done
3. Lane 109: the word “was” is missing.
Response 1: Done
4. Lines 114-116: please clarify this sentence.
Response 1: Done
5. Table 1: “e” of “mouse” is missing.
Response 1: Done
6. Figure 13: please add “Striatal (?) neurons in the brain of…”
Response 1: Done
7. Lanes 506-507: The presence of PrPsc in follicular dendritic cells is also true for other TSEs.
Response 1: Done